# Reforming the Barrier: The Role of Formins in Wound Repair

**DOI:** 10.3390/cells11182779

**Published:** 2022-09-06

**Authors:** Parinaz Ahangar, Allison J. Cowin

**Affiliations:** Future Industries Institute, UniSA STEM, University of South Australia, Adelaide, SA 5000, Australia

**Keywords:** formins, wound healing, actin filaments, microtubules, inflammation, proliferation, migration

## Abstract

The restoration of an intact epidermal barrier after wound injury is the culmination of a highly complex and exquisitely regulated physiological process involving multiple cells and tissues, overlapping dynamic events and protein synthesis and regulation. Central to this process is the cytoskeleton, a system of intracellular proteins that are instrumental in regulating important processes involved in wound repair including chemotaxis, cytokinesis, proliferation, migration, and phagocytosis. One highly conserved family of cytoskeletal proteins that are emerging as major regulators of actin and microtubule nucleation, polymerization, and stabilization are the formins. The formin family includes 15 different proteins categorized into seven subfamilies based on three formin homology domains (FH1, FH2, and FH3). The formins themselves are regulated in different ways including autoinhibition, activation, and localization by a range of proteins, including Rho GTPases. Herein, we describe the roles and effects of the formin family of cytoskeletal proteins on the fundamental process of wound healing and highlight recent advances relating to their important functions, mechanisms, and regulation at the molecular and cellular levels.

## 1. Introduction

Wound healing is a finely tuned, multistep process that includes three main stages: inflammation, proliferation, and maturation [1]. This process comprises the spatial and temporal orchestration of different cell types that interplay with cytokines, chemokines secreted to the extracellular space by other cells [2]. Similar to other physiological processes, cells require a change in their shape and arrangement in every stage of the healing process that is mediated through cytoskeleton filaments and their binding proteins [3]. The cell cytoskeleton, which includes actin networks and microtubules, regulates the wound repair process by providing the mechanical support required for polarity, proliferation, migration, differentiation, and function of different types of cells in the wound bed [4,5,6].

Actin binding proteins regulate every aspect of actin dynamics including actin nucleation, elongation, polymerization, severity, and the cross-linking of the actin filaments [7]. Microtubule-associated proteins regulate the microtubule’s structure, function, and dynamic instability [8]. Overall, actin- and microtubule-associated proteins control the rate and extent of the assembly and disassembly of these filaments.

One group of actin- and microtubule-binding proteins that affect many cellular functions known to be important to wound repair is the formin family. Due to their effect on actin and microtubule nucleation, polymerization, and stabilization, formins regulate important wound-related processes including cell polarity, morphogenesis, chemotaxis, cytokinesis, proliferation, migration, and phagocytosis [9]. Herein, we describe the roles and effects of the formin family on the fundamental process of wound healing and highlight recent advances relating to their important functions, mechanisms, and regulation at the molecular and cellular levels.

## 2. Formins, DRFs, and Their Structure

Formins are a large (>1000 aa) and highly conserved protein family including 15 different proteins in vertebrates with a large diversity in their ability to nucleate actin filaments and regulate actin rearrangements [10]. Another fundamental role of formins is their dynamic regulation of microtubules [11,12]. Most formin proteins have three formin homology domains (FH1, FH2, and FH3), but just the first two (FH1 and FH2) are present in all formins [13]. A large number of proteins in the formin family are categorized into seven subfamilies based on the location of their FH2 domains: 1. Dia proteins (Dia1, Dia2, Dia3) or Diaphanous homolog formins (DIAPH1, DIAPH2, DIAPH3), 2. disheveled associated activator of morphogenesis (DAAM1 and DAAM2), 3. formins (FMN1, FMN2), 4. formin-like (FMNL1, FMNL2, FMNL3), 5. formin homology domain-containing protein (FHOD), 6. inverted, and 7. Grid2-interacting protein (GRID2IP) [14]. DIAPH, DAAM, FMNL, and FHOD are known as the Diaphanous-related formins (DRFs) [15].

## 3. Formins Regulation

DRFs are autoregulatory proteins that regulate their own function by a common modular architecture. These proteins possess an N-terminal GTPase-binding domain (GBD), which is activated by Rho GTPases, followed by a diaphanous inhibitory domain (DID) and a dimerization domain (DD) [16]. There is a diaphanous autoregulatory domain (DAD) in the C-terminal (Figure 1a); the DID domain binds to the DAD domain in a closed state, leading to an autoinhibited conformation of the formin [16]. DRFs are mainly regulated by Rho GTPases that bind to GBD and DID domains and disrupt the autoregulatory complex [17] (Figure 1b). In addition to Rho GTPases, the cytoskeletal protein Flightless I has been shown to regulate the actin assembly effect of DRFs [18]. Flightless I is an actin regulatory protein [19,20] that directly binds to mDia1 and DAAM1 and cooperates with Rho GTPase to block the autoinhibitory effect of DRFs [18]. This activation of DRFs by Flightless I elevates the actin assembly activity of FH1FH2 domains, consequently promoting the formation of actin-rich structures [18] (Figure 1b). In some cases, the localization and specific targeting of formins are regulated through posttranslational modification such as phosphorylation, myristoylation, and farnesylation [9,21,22].

## 4. Formins’ Contribution to Action Nucleation and Elongation 

Cells require actin nucleating and elongation factors to mediate the de novo assembly of actin filaments and regulate the rate and extent of actin polymerization [23]. Formins are the initiator of actin nucleation and elongation that mediate actin assembly and reorganization in unbranched actin filaments (Figure 2a), whereas branched actin networks are assembled mainly by the Arp2/3 complex coupled with the Wiskott–Aldrich syndrome protein family (WASP) [24]. The actin nucleation and polymerization activities of formins are mediated by their FH domains. The FH2 domain of formin proteins promotes the nucleation of unbranched actin filaments from free actin monomers (not profilin-actins) in bulk kinetic assays [25]. The stabilization of actin dimers has been proposed as the possible mechanism of FH2-stimulated actin nucleation [26,27].

FH2 and FH1 domains are known to mediate formin’s role in actin elongation. During elongation, the FH2 domain forms a head-to-tail ring-like dimer that surrounds the barbed end of the actin filament that allows the addition of tens of thousands of actin monomers [28]. The continuous binding of FH2 to the barbed end of F-actin increases the stability of the actin filament. In the absence of ATP hydrolysis, the elongation is very slow due to a lack of energy and a low number of actin monomers in the barbed ends [29]. The presence of the FH1 domain increases the rate of elongation by binding to the profilin–actin complex, leading to the hydrolysis of ATP coupled in the complex leading to an increase in the addition of actin monomers (Figure 2a) [29,30].

## 5. Formins’ Contribution to Microtubule Dynamics 

Formins have been recognized as one of the main regulatory factors of microtubule dynamics, especially during interphase and mitosis [31]. It has been shown that formins stabilize microtubules in migrating cells [32]. Dynamic microtubules grow and shrink constantly at their plus end; however, stable microtubules plus end faces the plasma membrane acting as a scaffold for motor proteins to transfer cargos required for cell dynamics [33]. Microtubule stabilization is crucial for a change in cell morphology, cell polarization, and directed migration. Formins also mediate the crosstalk between actin filaments and microtubules [34]. Microtubules link to formins through their FH2 domain, except for Formin 1, which interacts with microtubules through a domain near the N-terminus (Figure 2b) [35]. In addition, formins also contain binding sites for microtubule plus-end-tracking proteins (+TIPs), which mediate the indirect binding of formins to microtubules (Figure 2b) [36]. Most of the current knowledge of this interaction has come from in vitro studies using cultured fibroblasts as in vivo studies are challenging due to the involvement of the FH2 domain in both actin- and microtubule-binding [36]. A study using cultured fibroblasts has shown that FH1FH2 mDia stabilizes microtubules against depolymerization and tubulin dissociation [34], and the addition of mDia1 and mDia2 to cells increases the number of stable microtubules [31,34]. Additional studies have shown that Formin 1 plays a role in the stabilization and localization of microtubules in Sertoli cells [37] following plus-end stabilization or capping. The signaling pathway has been shown in another study on wounded serum-starved NIH-3T3 fibroblasts, which has identified Rho GTPases as the main upstream regulator of formins that release formin’s autoinhibition [15].

mDia has independent binding sites for microtubule binding protein (EB1) and adenomatous polyposis coli (APC), which leads to the formation of the mDia–APC–EB1 complex. APC and EB1 function downstream of the Rho–mDia signaling pathway and capture the positive end of microtubules, leading to microtubule stabilization [38].

Microtubules assemble into arrays of filaments called microtubule bundles, structures that are required for cell division and cytokinesis [39]. Formins are also able to bundle microtubules in vitro or bundle them with actin filaments via their FH1FH2 domain [34].

## 6. Formins’ Involvement in Wound-Healing-Related Processes

Cytoskeleton structure including actin filaments and microtubules is one of the main regulators of physiological processes that are fundamental for wound repair [3]. Hemostasis and inflammation are immediately activated following injury, to control bleeding and infection. Dead cells and pathogens are removed by neutrophils and macrophages that also release cytokines to promote the inflammatory response. Reepithelialization results in the restoration of an intact barrier during the proliferative stage of healing, which also includes the deposition of a new extracellular matrix and the revascularization of the newly formed granulation tissue. All of these processes are reliant on the proliferation, migration, and synthetic activities of keratinocytes, fibroblasts, and endothelial cells to restore the damaged tissue. Lastly, granulation tissue is remodeled via a process of protein degradation and synthesis to form functional skin [40,41]. These processes are all controlled at a sub-cellular level by the actin cytoskeleton and its binding proteins and are described in more detail below.

### 6.1. Formins in Inflammation

Formins are involved in the regulation of the inflammation phase of healing through their fundamental role in controlling cell polarity, dynamics, and the migration of inflammatory cells (Figure 3). Neutrophils are the first inflammatory cells that migrate towards the wound bed from blood vessels, and actin reorganization plays an essential role in neutrophil chemotaxis [42]. The predominant actin nucleating proteins found in neutrophils are Dia proteins [43,44]. The deletion of mDia1 has been shown to impair neutrophil polarization and directed migration, a function that was found to be associated with WASP at the leading edge of these cells [43]. Leukocytes such as neutrophils, B and T cells, and monocytes migrate to the wound site from blood vessels, a process called transendothelial migration (TEM) [45]. TEM is a multi-step process that includes the capture of leukocytes on endothelial cells by rolling, crawling, and the adhesion of leukocytes on an endothelial monolayer and eventually migrating over the monolayer [45]. The depletion of mDia1 impairs the ability of cells to undergo TEM [46].

Macrophages are additional important inflammatory cells that are recruited to the wound to remove pathogens and debris through their phagocytosis ability [47]. Macrophages have actin-rich protrusions, called podosomes, which are adhesion structures that facilitate tissue invasion and macrophage movements through complex tissues for immune surveillance [48]. The phagocytic uptake of antigens by macrophages depends on the polymerization of actin filaments [49]. Formins (FMNL1) are the main regulator of actin reorganization in podosomes, and any reduction in FMNL1 activity disrupts podosome structures [50]. Macrophage phagocytosis also relies on the activity of formins (mDia1, mDia2, FMNL1) [51,52], which are enriched at macrophage pseudopodia and regulate actin re-organization in the phagocytic cup during complement receptor (CR3)-mediated phagocytosis as a downstream effector of RhoA–ROCK signaling [53,54].

The migration and entry of lymphocytes, including B and T cells, to damaged tissue, is essential for the adaptive immune response of the body. Impaired T cell trafficking is observed in FMNL1 knock-out mouse models, which were shown to have inflamed tissues, indicating the important role of formins in T-cell morphology and mobility. This role is likely due to formin’s function in actin nucleation and polymerization at the back of migrating T cells [55]. Indeed, T cells of mDia1−/− mice have reduced actin polymerization in vitro, and T cell trafficking is disrupted and inefficient in vivo [56]. Diminished T cell populations in lymphoid tissues have also been observed in DRF1−/− mice. Isolated T cells from the spleen of DRF1−/− mice were less adhesive to the extracellular matrix and showed impaired migration [57]. mDia1−/− mice also have impaired adhesion and spread to the cellular matrix in dendritic cells. Furthermore, T-cell stimulation is also impaired in these mice [58].

Formins are important regulators of the T cell synapse. Actin assembly and cytoskeleton rearrangement are involved in immunological synapses [59]. mDia1 and FMNL1 have been found to be localized in the lamellipodium of T cells, forming the immunological synapse [60]. These formins also regulate MTOC polarization in T cells when they encounter an antigen-presenting cell (APC) in immunological synapses [60].

Natural killer cells are innate immune cells that fight bacteria and have a role in the resolution of inflammation during wound healing [61]. Formins facilitate cell adhesion, signaling, and chemotaxis in natural killer cells and have been shown to regulate microtubules and promote the development and polarization of the cytolytic granules of natural killer cells [62,63].

### 6.2. Formins in Skin Cell Migration

Cell migration relies on the reorganization of the actin cytoskeleton into complex actin-rich structures, such as filopodia and lamellipodia, at the front edge of migrating cells [3]. These thin protrusive extensions are required for directed migration, exploring the extracellular matrix, and penetrating tissue spaces. They are also well suited for intercalating between cells, such as during the migration of leukocytes across endothelial layers [64]. Formins are involved in filopodia formation, which is a highly dynamic process creating thin protrusions that are rich in parallel unbranched actin filaments [65]. The extension of these protrusions occurs by the elongation and capping of the barbed ends of actin filaments [66]. Rho GTPase family proteins are known to be the main regulator of filopodia formation and rearrangements [67]. Rho GTPase blocks the autoinhibitory switch of DRFs, and active DRFs are able to nucleate actin filaments [29] and cap the barbed ends by their FH2 domain, stabilizing the formation of an adjacent actin dimer [68]. The capping action allows the actin nucleus to elongate from its barbed end [29,69]. The role of formins in filopodia formation can be disrupted by the interaction between formins and either the WASP or Arp2/3 complex, which strikes a balance between the formation of filopodia and lamellipodia in migrating cells [67,70].

Lamellipodia are actin-rich protrusions that are composed of a branched actin filament meshwork assembled by the Arp2/3 complex and the WASP family [71]. However, several studies show an important role for formins (mainly mDia2) in the formation of lamellipodia, which is mediated by nucleating the actin filaments and protecting them from capping [71,72].

In addition to actin filaments, formins (mainly mDia1 and mDia2) regulate microtubule dynamics, which is essential for cell polarity and directed migration. Not only do formins bind and stabilize the microtubules [73], but mDia1 has been found to polarize microtubules from the cell center microtubule-organizing center (MTOC) to the periphery in migrating cells towards the direction of cell migration [74,75]. Furthermore, mDia2 stabilizes microtubules, which is essential for cell migration to occur [34].

The formation of focal adhesions, as well as their turnover on the cell surface, allows connections to occur between cells, basal tissue, and the extracellular environment [76]. Extracellular stimuli (such as physical or chemical) are recognized by cells via their focal adhesions, leading to appropriate cellular responses. During migration, alterations in focal adhesions allow cells to respond to the environment by cytoskeletal rearrangement and the promotion of cell protrusions. Focal adhesions are also used by migrating cells as cortical anchors: an elongation spot for cytoskeleton protrusions [77]. Several studies have demonstrated the effect of formins on promoting the formation and turnover of focal adhesions. mDia1 knockdown resulted in the inhibition of focal adhesion formation in SKBR3 and T47D breast carcinoma cells [75]. The depletion of mDia1 protein in cancer cells also resulted in a lower amount of Src (a family of protein tyrosine kinases) accumulation into focal adhesions leading to impaired focal adhesion functionality and stability and impaired cell migration [78]. mDia1 and FMNL3 formins have also been discovered to localize adherent junctions in epithelial cells and maintain the monolayer integrity of these cells during wound healing [79]. These formins accelerate actin polymerization at these junctions, and their depletion reduced the stability of E-cadherin and the disruption of cell–cell adhesion [79]. Finally, the overexpression of FHOD1 has been shown to induce cell elongation, and migration and alteration in FHOD1′s structure have resulted in impaired cell migration without affecting adhesion [80].

### 6.3. Formins in Cell Proliferation

To replace damaged tissue during wound healing, cell proliferation occurs, leading to the restoration of the epidermis and the formation of new dermis through the production of granulation tissue [81]. Cytokinesis is the final stage of cell division that divides the cytoplasm of a cell into two cells following mitosis. Cytokinesis begins with the assembly of an actomyosin-rich contractile ring. When the ring contracts, a cleavage furrow forms, which eventually separates the two sides of the ring [82]. The positioning and induction of a cleavage furrow on the metaphase plate are regulated by microtubules [83]. Clusters of recycled endosomes are required as the main source of the additional membrane around the cleavage furrow to increase cell surface area and the accommodation of the cell shape and polarity changes during cytokinesis [84]. The accumulation of endosomes occurs in the midbody area near MTOC, which relies upon activity of microtubule motors [85].

Formins play a critical role in cytokinesis. Several studies have shown the failure of cytokinesis following a mutation or the genetic deletion of formin proteins. There are different functions of formin proteins during cytokinesis, which rely on their role in actin reorganization and microtubule stabilization. Rho-regulated DRFs are the main stimulator of actin assembly in the contractile cortex [86]. Formins have been identified as essential factors for the formation/activity of contractile rings during cytokinesis in drosophila [87,88], C. elegans [89], yeasts [90], and mammals [91]. DRFs localize in pericentrosomal dividing cells near the contractile ring and furrow, which is mediated by Rho-GTPase regulation [92]. mDia1 and mDia2 localize to the microtubules of proliferating cells and facilitate cytokinesis by stabilizing the microtubules [93]. Formins are also known to act as a link between microtubules and actin filaments during cytokinesis and regulate their positioning. The overexpression of formins leads to the disruption of the alignment of microtubules and actin filaments [94]. A study on fibroblasts and Xenopus embryos showed that the Rhod–hDia2C–Src pathway involves the interaction of endosomal vesicles with microtubules and actin [95]. In addition, RhoA-activated DRFs are involved in the stability of the cytokinesis furrow by assembling β-actin filaments at the site of cytokinesis and directly at the furrow [96].

### 6.4. Formins in Epithelial-to-Mesenchymal Transition (EMT)

Epithelial-to-mesenchymal transition (EMT) is a vital part of the wound-healing process that occurs during re-epithelization and is mediated by inflammatory cells and fibroblasts [97]. During re-epithelization, keratinocytes proliferate and migrate to restore the epithelial barrier. Re-epithelialization is supported by the conversion of cells from a stationary state to a migratory one, mediated by EMT [98]. Keratinocytes go through cytoskeleton rearrangement, lose their polarity and cell–cell adhesions, modulate their interaction with the ECM, and obtain mesenchymal features [99]. The cytoskeleton rearrangement during EMT is regulated by transforming growth factor β1 (TGF-β1) and its downstream effectors RhoA GTPase and formins including DIAPH1 and DIAPH3 FHOD1 and FMNL2 [100,101,102].

### 6.5. Formins in Angiogenesis

The process of new blood vessel formation, called angiogenesis, is critical for effective wound healing. Following the resolution of inflammation, newly branched blood vessels invade granulation tissue to provide nutrition and oxygen to the newly formed tissue [103]. Actin reorganization is required for endothelial cell (EC) polarization, proliferation, migration, and adherence [3]. Formins are one of the cytoskeletal regulators of angiogenesis with conflicting effects on this process. Due to formin’s role in actin polymerization, these proteins, especially FMNL3, are required for filopodia formation in migrating endothelial cells during angiogenesis [104]. Several studies show an extending role of formins beyond their effect on EC migration. ECs are highly flattened cells in a quiescent state. However, during angiogenesis, these cells undergo a change in their morphology and polarity in order to initiate migration [105]. EC microtubules realign and stabilize during angiogenesis, leading to a change in the morphology of ECs. Formin screening has identified FMNL3 as the regulator of angiogenic ECs′ morphogenesis, and silencing FMNL3 has led to the inhibition of blood vessel formation. FMNL3 has been shown to act as a downstream effector of Cdc42 and RhoJ and regulates microtubule alignment during EC morphogenesis [106,107]. During angiogenesis, ECs establish cell–cell junctions and rearrange for new vessel formation and stabilization. Actin filaments polymerize and assemble in these EC junctions. FMNL3 has been shown to localize in EC junctions where they promote actin filament polymerization. FMNL3 knockdown has also led to impaired actin filaments′ polymerization and stabilization highlighting the importance of this formin in maintaining EC junctions [108].

DAAM1 formin has been identified as a promoting factor of both the microtubule stabilization and actin polymerization of ECs. However, the overexpression of DAAM1 elevates microtubule stabilization rather than actin polymerization and inhibits angiogenesis by inhibiting the proliferation and migration of ECs [109].

Growth factors such as vascular endothelial growth factor (VEGF) and angiopoietin-1 (Ang-1) promote angiogenesis through two different signaling pathways. VEGF increases endothelial cell permeability through the activation of Src kinase via its receptor VEGFR. Ang-1 inhibits VEGF-stimulated permeability by activating RhoA and its downstream effector mDia [110,111]. mDia has been shown to interact directly with Src and inhibits its activity by sequestering it from the VEGFR pool, which ultimately blocks EC permeability and promotes barrier integrity. This function of Ang-1 is important in protecting blood vessels in chronic wounds with persistent inflammation [111].

### 6.6. Formins in Tissue Maturation and Fibrosis

Maturation is the final stage of wound healing, which involves the remodeling of granulation tissue. Myofibroblast differentiation is a crucial aspect of this process as they promote wound contraction and the realignment of ECM components, which restores tissue integrity. Fibroblasts differentiate into myofibroblasts under the mechanical stress of the ECM. During this transition, myofibroblasts develop highly organized and contractile actin filament bundles called stress fibers [112]. Stress fibers are assembled by formin (mDia1/mDia2)-driven actin polymerization at focal adhesions, and silencing mDia1/2 in stress fibers disrupts myofibroblast differentiation [113]. One of the main regulators of myofibroblast differentiation is TGF-β1, which promotes the formation of stress fibers and giant focal adhesions in myofibroblasts [114]. mDia proteins are recruited to stress fibers via TGF-β1/RhoA signaling and facilitate myofibroblast differentiation by promoting actin filament polymerization (Figure 4). In addition, myofibroblast differentiation is regulated by the interactions of the microtubule system and actin cytoskeleton via mechanical coupling. TGF-β1 signaling is blocked by microtubule polymerization, preventing myofibroblast differentiation. mDia2 interacts with microtubules and its localization to stress fibers can be regulated by microtubules [115].

## 7. Pharmacological Approaches to Modulate Formins

The pharmacological modulation of actin-related proteins and regulating factors is a valuable tool to assess the mechanism of actions of these factors and highlight their potential use as targeted therapies [116]. A general small-molecule inhibitor of formin homology 2 domains (SMIFH2) has been identified by Rizvi et al., using the in vitro screening of different compounds for formin inhibition, which was found to be active against several types of formins from different species [117]. SMIFH2 has been widely used in most recent studies to evaluate the role of formin in different biological processes. It has been shown to inhibit formin-dependent actin polymerization, reduce cell proliferation, and disrupt cytokinesis and microtubule dynamics [118]. The blocking effect of SMIFH2 on formins has been shown to perturb the actin cytoskeleton through the remodeling of actin filaments and microtubules [119]. This molecule disrupts actin filament relocation and stress fiber contraction. It also inhibits non-muscle myosin 2A and skeletal muscle myosin 2 in vitro and decreases myosin ATPase activity [120]. The treatment of human and murine platelets with SMIFH2 resulted in a decrease in the cell size and spread due to the disorganization and dynamics of actin and microtubules that point out to the role of formin in cross-talk between the actin and microtubule filaments [121]. As mentioned above, SMIFH2 has been shown to affect myosin proteins in addition to formins, which indicates its off-target activity. Therefore, all data acquired by this inhibitor should be interpreted with caution [122]. Since formins are heavily involved in wound-repair-related processes, more pharmacological research is required to modulate them in order to investigate their role in wound healing in vitro and in vivo.

## 8. Conclusions

Formins have been identified as effector proteins in many of the cellular processes that are critical for successful wound healing. They regulate inflammatory cell recruitment to the wound bed and promote cell proliferation and migration, as well as accelerate new blood vessel formation. They support the differentiation of fibroblasts to fibrosis-related myofibroblasts and facilitate the formation of new tissue. However, there are also still many important questions that remain to be answered from a structural and mechanistic standpoint about the role of formins in wound healing. There is still a need to develop new formin inhibitors that inhibit the different domains of these proteins to specifically understand the molecular mechanisms that these proteins are involved in. To date, limited studies have investigated the role of formins during wound healing per se, and even fewer studies have investigated their function in wounds that fail to heal such as in diabetic foot ulcers, chronic venous leg ulcers, or burn injury repair. This gap in knowledge of the role of formins at a physiological level highlights a need for research in this area. Understanding the function of formins during impaired wound healing may help to identify molecular approaches for stimulating hard-to-heal wound responses.

## Figures and Tables

**Figure 1 cells-11-02779-f001:**
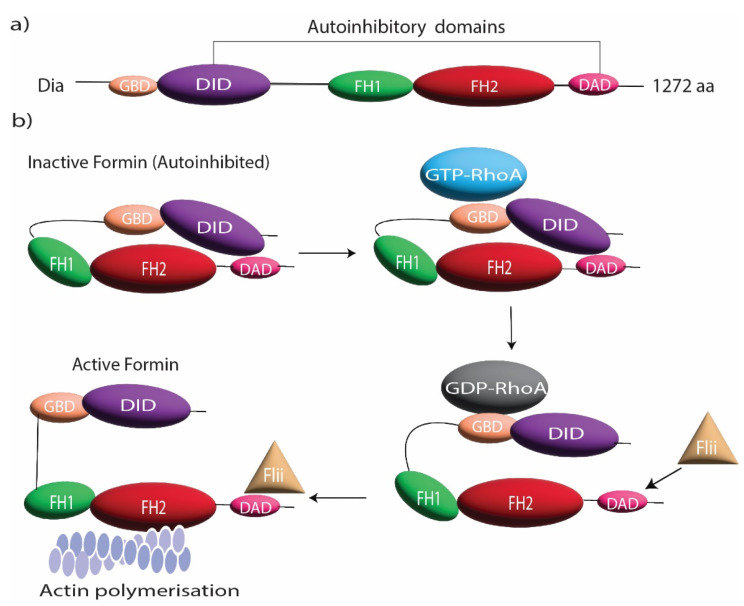
Formin structure and regulation. (**a**) Domain organization of DRFs including GBD, DID, FH1, FH2, and DAD. (**b**) The domain organization during the autoinhibitory regulation of DRFs and its release by RhoGTPase and Flightless I (Flii).

**Figure 2 cells-11-02779-f002:**
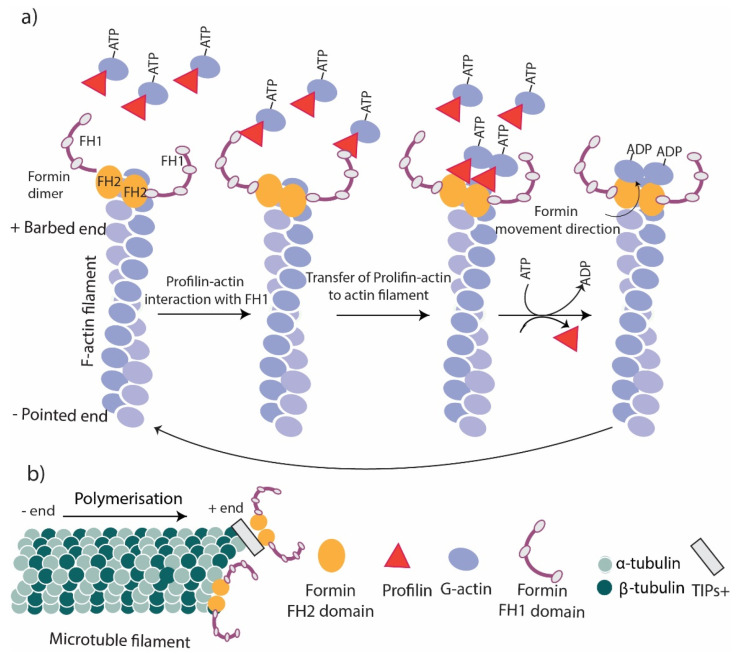
Formins function. (**a**) Actin polymerization. The FH2 domain of formins remains at the barbed end of actin filaments to stabilize and cap the actin filament. The interaction of FH2 with the FH1/profilin complex transfers new G actins to the barbed end of the filament and promotes polymerization. (**b**) Microtubule stabilization. Formins link to the + end of microtubules directly or via their FH2 domain or through TIPs.

**Figure 3 cells-11-02779-f003:**
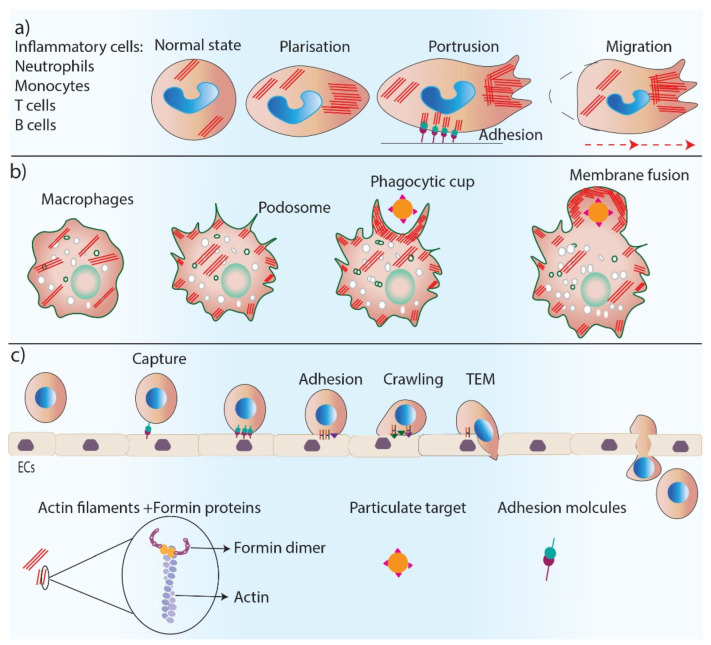
Formins’ role in inflammation. Formins play an essential role in inflammation by regulating actin polymerization in inflammatory cells. (**a**) Formins control cell polarity, cell protrusions and directed migration in inflammatory cells. (**b**) Formins are involved in podosome and phagocytic cup formation and the phagocytosis process. (**c**) Trans-endothelial migration of inflammatory cells requires several steps including, cell capture, adhesion, crawling and TEM which are all regulated by formin-mediated actin polymerization.

**Figure 4 cells-11-02779-f004:**
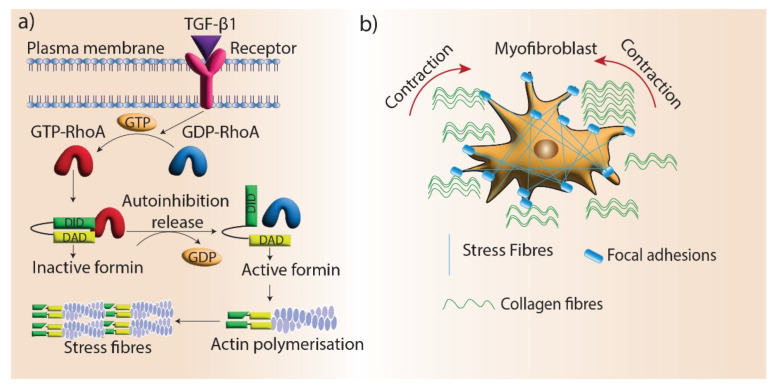
Formins’ role in myofibroblast contraction. (**a**) Stress fiber generation by formin-mediated actin assembly. TGF-β1 signaling promotes stress fiber formation in myofibroblasts by mediating the activation of RhoGTPase, which blocks the autoinhibition of formins and promotes actin assembly. (**b**) Stress fibers locate in the focal adhesions of myofibroblasts and facilitate the contraction and maturation of new ECM.

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
