# Peer review of "Reforming the Barrier: The Role of Formins in Wound Repair"

_cells, 2022, doi:10.3390/cells11182779_

Round 1

Reviewer 1 Report

This interesting review paper describes the roles of formin proteins in the regulation of wound healing. The authors are leading expert in the field that published a number of impactful studies on the cytoskeletal mechanisms of tissue injury and repair. The review is logical and well-written. The reference list is up to date and comprehensive. I have just a few minor suggestions.

1. The author's list has an accidental 'and' at the end.

2. Lines 56-60. The formin classification is a bit confusing. Diapharnous related formins, are they just DIAPH1, DIAPH2 and DIAPH3, or DIAPH, DAAM, FMNL and FHOD altogether?  Furthrmore, FHOD should be de-abbreviated.

3.  lines 158-164: The authors described the roles of firmins in regulating leukocyte migration during inflammation, however, this is just a part of the story. I wonder if anything is known about formins regulating interactions between leukocytes and vascular endothelium , which is critical for leukocyte transmigration into inflamed tissues.

4. In regard to leukocyte functions, are there any roles of formins in formation of immunological synapse between T cells and antigen-presenting cells? Such interactions are known to depend on actin cytoskeleton.

5.  Section 6.2 should be entitled "Formins in epithelial cell migration' since immune cell migration was described in the previous section.

6. Section 6.5 could be better defined as 'Formins in tissue maturation and fibrosis'

7. Lines 346-347.  The authors should more vividly emphasize possible off-target effects of a common formin inhibitor, SMIFH2, and state that the obtained results with this inhibitor should be interpreted with caution.

Author Response

  1. The author's list has an accidental 'and' at the end.

Thank you, we have corrected this mistake (P1, Line 3).

  1. Lines 56-60. The formin classification is a bit confusing. Diapharnous related formins, are they just DIAPH1, DIAPH2 and DIAPH3, or DIAPH, DAAM, FMNL and FHOD altogether?  Furthrmore, FHOD should be de-abbreviated.

Thank you for your comment. We have modified the text to avoid confusion (P2, Lines 56-60).

  1. lines 158-164: The authors described the roles of firmins in regulating leukocyte migration during inflammation, however, this is just a part of the story. I wonder if anything is known about formins regulating interactions between leukocytes and vascular endothelium , which is critical for leukocyte transmigration into inflamed tissues.

Thank you for this suggestion. We have already included studies on the impact of Formins during transendothelial migration. We could not find any more studies in the literature (P5, Lines 164-165).

  1. In regard to leukocyte functions, are there any roles of formins in formation of immunological synapse between T cells and antigen-presenting cells? Such interactions are known to depend on actin cytoskeleton.

Thank you for the suggestion. We have now added a section about the effect of formins in immunological synapses (P6, Lines 189-193).

  1. Section 6.2 should be entitled "Formins in epithelial cell migration' since immune cell migration was described in the previous section.

We have made this change to the section heading, please see P6, Line 206.

  1. Section 6.5 could be better defined as 'Formins in tissue maturation and fibrosis'

We have made this change to the section heading, please see P9, Line 327.

  1. Lines 346-347.  The authors should more vividly emphasize possible off-target effects of a common formin inhibitor, SMIFH2, and state that the obtained results with this inhibitor should be interpreted with caution.

Thank you for the suggestion. We have included a statement regarding the limitation of this inhibitor (P10, Lines 366-369).

Reviewer 2 Report

The present review paper is weel written and designed. the topic is interesting and innovative.

I suggest to the authors to add a paragraph on the future perspectives and add the limitations of the review in the conclusion paragraph.

Moreover I also suggest to add some informations on the wound healing and the related molecular mechanisms (fpr e.g. DOI: 10.3390/cells10071587,  doi: 10.3389/fphys.2021.676512).

Author Response

  1. I suggest to the authors to add a paragraph on the future perspectives and add the limitations of the review in the conclusion paragraph.

Thank you for the suggestion. We have added limitations and perspectives to the conclusion paragraph (P10, Lines 377-385)

  1. Moreover I also suggest to add some informations on the wound healing and the related molecular mechanisms (fpr e.g. DOI: 3390/cells10071587,  doi: 10.3389/fphys.2021.676512).

Thank you for your comments and suggestions. We have now included a section to include information on wound healing about the effect of formin on EMT (P8, Lines 283-292).

Reviewer 3 Report

In my opinion the paper is an excellent and very well-written review. I like very much 4 figures illustrating the role of formins and the mechanisms of their functioning.

Minor comment

Fig 3a please correct plarisation -> polarisation

Author Response

Fig 3a please correct plarisation -> polarisation

Thank you for spotting this mistake. We have modified accordingly (New figure 3).